**Data Availability Statement:** Mobility data are collected from PlaceIQ (https://www.placeiq.com/), Google Mobility (https://www.google.com/covid19/mobility/), and SafeGraph (freely provided upon request at https://www.safegraph.com/covid-19-

# Financial relief policy and social distancing during the COVID-19 pandemic

Cody Yu-Ling Hsiao [1,2☯], Stanley Iat-Meng Ko [3,4,5☯], Nan Zhou [6☯]*

**1** School of Business, Macau University of Science and Technology, Cotai, Macau, **2** Centre for Applied Macroeconomic Analysis, Australian National University, Canberra, Australia, **3** Graduate School of Economics and Management, Tohoku University, Sendai, Japan, **4** Center for Data Science and Service Research, Tohoku University, Sendai, Japan, **5** Policy Design Lab, Tohoku University, Sendai, Japan, **6** Krieger School of Arts and Sciences, Johns Hopkins University, Baltimore, MD, United States of America

☯ These authors contributed equally to this work.

* nzhou4@jh.edu

## Abstract

In this paper, we investigate the effect of stimulus payments during the COVID-19 pandemic on the social distancing practices of their recipients. While the directed cash payments stipulated by the 2020 CARES Act were intended to mitigate the economic impact of closures imposed in response to the outbreak, we find that this relief may also have inadvertently contributed to the spread of the virus due to increased social activity. We find that, as the payments were sent out on a staggered weekly schedule, there was a corresponding spike in weekend traffic as indicated by a number of mobility metrics that measure social distancing, over and above the usual uptick expected from weekend shopping following receipt of the stimulus payments on Fridays. This preliminary study gives some indication that the economic benefits of the stimulus package may in fact be outweighed by the detrimental effects of looser social distancing practices prolonging the outbreak.

## Introduction

One of the major obstacles faced by all governments during the COVID-19 pandemic was the question of how to limit the spread of the virus without inflicting significant harm to the economy. With the shelter at home orders during the early months of the pandemic leaving many businesses shuttered and individuals out of work, it quickly became evident that some type of financial relief was needed to help households meet obligations and to stimulate economic activity. As shown in Baker et al. (2020) [1], direct payments are not necessarily the most effective means to stimulate consumption, but following the precedent of 2001 and 2008, the CARES Act passed on March 25th, 2020 stipulated payments of $1,200 per adult and $500 per child, phasing out at higher household income levels.

However, a unique obstacle presented by the pandemic that served to undermine the efficacy of stimulus measures was that combating the disease required individuals to avoid contact with others so as to not spread the virus, and the term "social distancing" quickly became prominent in the colloquial lexicon. While online shopping had been growing even before the

data-consortium). Income quantiles were constructed using American Community Surveys (ACS) data (2014–2018, 5-year pooled) on the ACS website (https://www.census.gov/programs-surveys/acs).

**Funding:** The authors received no specific funding for this work.

**Competing interests:** The authors have declared that no competing interests exist.

pandemic, much of the economy that policymakers hoped to stimulate was still dependent on physical retail stores, particularly as these employed many more workers than online stores. Therefore, handing out stimulus checks would inevitably encourage recipients to go shopping and come into close contact with each other, potentially spreading the virus. From a broader view, this could even serve to unnecessarily prolong the pandemic and extend lockdown measures, which would negate much of the economic impact that the stimulus was intended to have.

Many epidemiologists have sounded alerts regarding the COVID-19 infection spread. Early literature shows evidence that reduced human mobility significantly reduces virus growth rates, as seen in Fang, Wang, and Yang (2020) [2] and Shibamoto, Hayaki, and Ogisu (2020) [3]. Additionally, a growing literature studies the impact of government-implemented policies on human behavior. Mendolia, Stavrunova, and Yerohkin (2021) [4] show that stringent policies such as restrictions on international travel and closures of schools and workplaces have negative and statistically significant effects on human mobility. To complement previous studies focusing on the government-imposed behavioral policies, our study further investigates the impact of financial relief policy on human mobility.

In this paper, we study the effect of stimulus payments, disbursed weekly starting on April 17th 2020, on the social distancing practices of recipients, using a variety of mobility measures. We find that in each case, the payments reduced social distancing and furthermore that this effect was differentiated by income level, a phenomenon explored in Weill et al. (2020) [5]. This result is especially surprising in light of evidence from Coibon, Gorodnichenko, and Weber(2020) [6] that much of the stimulus went to pay down debt, indicating that even the relatively small proportion that did go to consumer spending had a significant impact on social distancing practice. This adds another dimension to studies of the overall effects of the pandemic response policies and the tradeoffs involved as in Kaplan, Moll, and Violante (2020) [7].

## Material and method

### Data

The data consists of the daily averages of four mobility measures in the United States from January 1st to June 30th, 2020. As shown in Fig 1, these four measures are averaged for each income quintile either by census tract (left) or county (right). Each panel shows a different daily mobility measure derived from mobile device location pings: i) percent of devices staying completely at home (top left), ii) device exposure, given by the average number of devices at all of the locations visited by a device in a day (top right), iii) median distance traveled outside the home, computed by taking the median distance traveled among the devices that left their home (bottom left), and iv) percentage change in device presence at locations of retail and recreation relative to the baseline day, set as the median value over the 5-week period from January 3th to February 6th, 2020 (bottom right).

All measures in Fig 1 show a significant increase in social distancing following state governments' emergency declarations in March. The social distancing measures continuously increase and peak in early April. When the financial relief policy was implemented in the middle of April, these measures started to decrease gradually. They also show a clear pattern of fluctuation over the course of the week, with less social distancing on weekends than on weekdays. This could be attributed to the stimulus payment being sent out on Fridays, triggering consumers to go shopping on weekends and in turn reducing social distancing. Besides this weekend effect, we also find that degree of social distancing differs with the income quintile, with the top quintile displaying the strongest distancing responses and the bottom quintile showing the weakest responses (The data provider SafeGraph uses multiple methods to

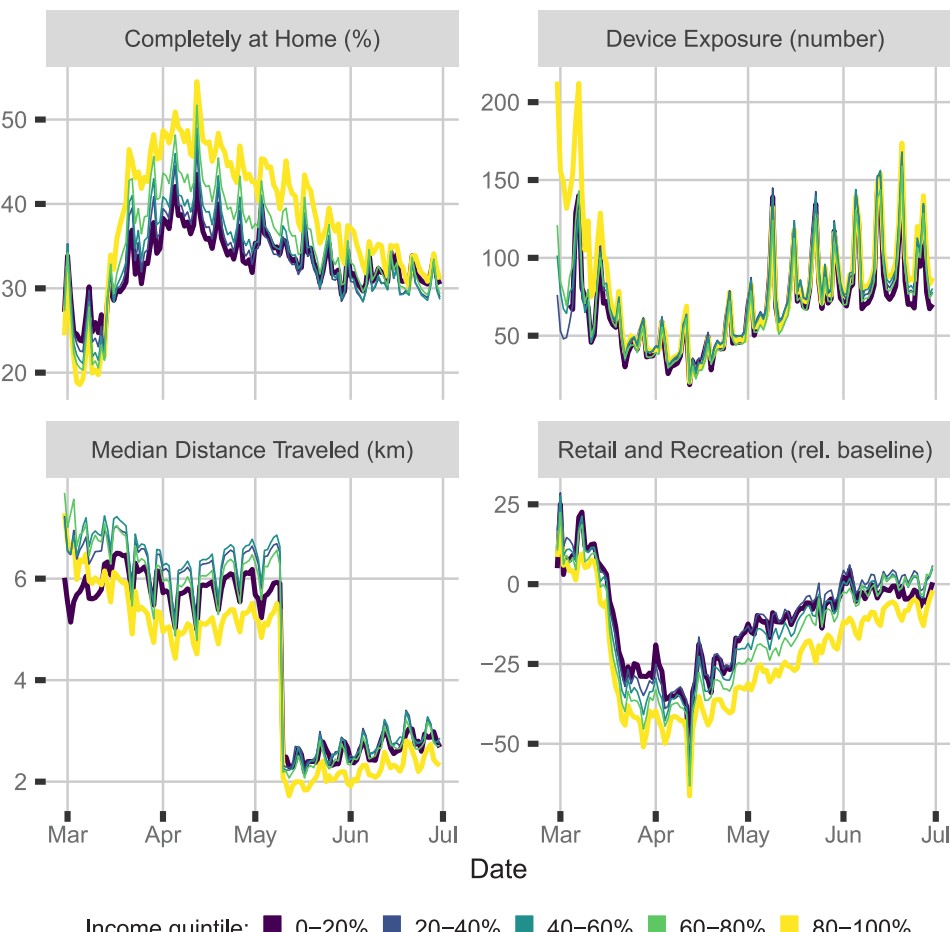

**Fig 1. Daily mean mobility measures in the United States from mobile devices.** Daily mean mobility measures in the United States from mobile devices starting from the declaration of emergency of each state in March to July 31, 2020 by quintiles of median income at the census tract (Left) or county (Right) level. Thicker lines indicate the top and bottom quintile. Each panel shows a different measure of social distancing behavior. Data are from SafeGraph, PlaceIQ, and Google.

measure "median distance traveled", so that there is a structural change in the series on May 9th due to the change of versions, explaining the sudden drop seen in the plot. See https://docs.safegraph.com/docs/social-distancing-metrics).

According to the CARES Act stimulus payments, taxpayers received the first payment on April 17th, 2020, using direct deposit information from their tax filings from 2018 or 2019, and over 80 million Americans received payments in their bank accounts. During the following weeks, the Internal Revenue Service (IRS) continued weekly rounds of direct deposits to those who provided direct deposit information through the IRS website. All other taxpayers who had not registered their bank account information by May 13th received their stimulus payment by paper check. In addition, starting from the week of April 24th, checks were mailed weekly according to gross income group, starting from the lowest with a gross income of less than 10k. Overall, 5 million checks were sent out per week until the end of August.

Fig 2 shows the periods of emergency declarations and CARES Act stimulus payments covered in our sample. A nationwide emergency for COVID-19 was declared on March 13th, 2020 following a similar declaration from the WHO on the 11th, although some states had

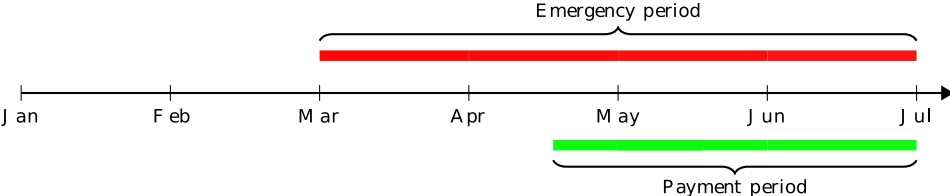

**Fig 2. Timeline of emergency declarations and CARES Act stimulus payments in our sample coverage in 2020.**

already declared an emergency earlier in March. The emergency status continued as the United States recorded 50,000 new cases of COVID-19 on July 2nd, the largest one-day spike since the pandemic's onset (See the timeline of COVID-19 developments in 2020 on the website of the American Journal of Managed Care https://www.ajmc.com/view/a-timeline-of-covid19-developments-in-2020). States postponed or reversed plans to reopen their economies in July. Thus, our sample period covers the most devastating pandemic period in the United States when the social distancing order was enforced. Stimulus payments began on April 17th, 2020, so our sample coverage balances the normal, emergency, and stimulus payment periods.

## Method

We propose the following regression model to further disentangle the payment, emergency declaration, and weekend effects on social distancing across the 5 income groups. Let $Y_{ct}$ be the mobility measure in county $c$ at time $t$, specifically Completely at Home (log), Median Distance Traveled (log), Device Exposure (log), or Retail and Recreation. The regression analyses are conducted with daily observations from January 2020 to June 2020 except for the median distance traveled measure where only the observations up to May 9th are included due to the inconsistency of data versions. We index income quintiles by $q \in Q = \{1, 2, \ldots, 5\}$. $D_q$ is income quintile $q$ dummy. $ED_{ct}$ is a dummy variable indicating the after state emergency declaration period, i.e., $ED_{ct} = 1$ when $t \geq$ state emergency declaration date in county $c$. $D_t^{wknd}$ is the weekend dummy variable for Saturday and Sunday. $D_t^{pay}$ indicates the period after the first stimulus payment, i.e. $D_t^{pay} = 1$ when $t \geq$ April 17th. Following Weill et al. (2020) [5], we also control for the cumulative number of COVID-19 infected cases in each county, $X_{ct}$. The regression model is thus

$$
\begin{aligned}
Y_{ct} \quad &= \sum_{q \in Q} c_q \cdot D_q + \sum_{q \in Q} \alpha_q \cdot D_q \times D_t^{wknd} + \sum_{q \in Q} \beta_q \cdot D_q \times ED_{ct} \\
&+ \sum_{q \in Q} \gamma_q \cdot D_q \times ED_{ct} \times D_t^{wknd} + \sum_{q \in Q} \theta_q \cdot D_q \times ED_{ct} \times D_t^{pay} \\
&+ \sum_{q \in Q} \eta_q \cdot D_q \times ED_{ct} \times D_t^{pay} \times D_t^{wknd} + \zeta \cdot X_{ct} + \epsilon_{ct}.
\end{aligned}
\tag{1}
$$

Model (1) is sometimes referred to as a saturated regression which utilizes dummies and their interactions to disentangle various marginal effects. Table 1 summarizes the marginal effects on mobility estimated from model (1). Since our sample provides balanced coverage of different episodes over the period, the sample size is sufficient to obtain consistent estimates.

**Table 1. Summary of marginal effects on mobility measures.**

| Regressor | Marginal effects estimated |
|---|---|
| $D_q$ | Average daily mobility of people in income quintile $q$ in *normal period* |
| $D_q \times D_t^{wknd}$ | Weekend extra mobility of people in income quintile $q$ in *normal period* |
| $D_q \times ED_{ct}$ | Daily extra mobility of people in income quintile $q$ in *emergency period* |
| $D_q \times ED_{ct} \times D_t^{wknd}$ | Weekend extra mobility of people in income quintile $q$ in *emergency period* |
| $D_q \times ED_{ct} \times D_t^{pay}$ | Daily extra mobility of people in income quintile $q$ *after the first stimulus payment* |
| $D_q \times ED_{ct} \times D_t^{pay} \times D_t^{wknd}$ | Weekend extra mobility of people in income quintile $q$ *after the first stimulus payment* |

Notes: $D_q$, $q = 1, \ldots, 5$, is the five income quintile dummies, $D_t^{wknd}$ is the weekend dummy, $ED_{ct}$ indicates the after state emergency declaration period in county $c$, and $D_t^{pay}$ indicates the stimulus payment period.

## Results

Table 2 presents regression models of four different mobility measures on income quintile ($D_q$), weekend effect ($D_t^{wknd}$), state emergency declaration ($ED_{ct}$), and stimulus payment period ($D_t^{pay}$). We found that, after the state of emergency declaration date ($ED_{ct} = 1$), social distancing increases significantly for all income groups. Moreover, social distancing responses range systematically from weakest for the bottom income quintile ($D_1 \times ED_{ct}$) to strongest for the top income quintile ($D_5 \times ED_{ct}$).

The state emergency declaration increased social distancing behavior on both weekdays and weekends by the three measures of device exposure, median distance traveled, and retail and recreation, but not by the proportion staying completely at home. In addition, the effect of the emergency declaration on weekends ($D_q \times ED_{ct} \times D_t^{wknd}$) is much smaller than on weekdays ($D_q \times ED_{ct}$).

After the stimulus payment on April 17th, social distancing behavior loosens for all income groups. Furthermore, by all four mobility measures, this drop in social distancing practices following the stimulus is also strongly differentiated by income quintile. Using the "completely at home" measure, the response to the stimulus payment is larger for the low income quintile ($D_2 \times ED_{ct} \times D_t^{pay}$), with 8.7% decrease, than for the high income quintile ($D_4 \times ED_{ct} \times D_t^{pay}$), with 6.4% decrease in weekdays. Substantially less social distancing for the low-income quintile is also apparent from "device exposure", which proxies how often people are going to crowded places. For "median distance traveled", the highest and lowest income travelers show greater increases in mobility after the stimulus payment, by 2.7% and 3% more on weekdays, than their middle-income counterparts. However, for the "retail and recreation" measure, middle-income counties show the greatest increases in weekday activity after the stimulus payment, indicating that both the highest and lowest income counties maintained better social distancing practices.

As the stimulus payment is disbursed on Friday, we are interested in analyzing the weekend effects of the stimulus policy on social distancing for each income quintile (the last panel in Table 2). We find that the social distancing practices become far worse on weekends than on weekdays after the stimulus payment for all income levels. Measured by "device exposure" and "median distance traveled", lower-income quintiles show an extra sharper reduction in social distancing on weekends in response to receiving their stimulus payments. By contrast, using the "retail and recreation" measure, middle-income group saw more activity on weekends following the payments compared to the highest and lowest income group.

**Table 2. The regression models of four separate mobility measures on income quintiles ($D_q$), weekend effect ($D_t^{wknd}$), state emergency declaration ($ED_{ct}$), and the stimulus payment period ($D_t^{pay}$).** Standard errors are given in parentheses.

| | Complete Home | Device Expos. | Med. Dist. Traveled | Retail & Recreation |
|---|---|---|---|---|
| $D_1$ | −1.583 | 4.433 | 9.019 | 8.718 |
| | (0.007) | (0.031) | (0.014) | (0.254) |
| $D_2$ | −1.550 | 4.453 | 8.959 | 8.932 |
| | (0.007) | (0.025) | (0.013) | (0.217) |
| $D_3$ | −1.527 | 4.500 | 8.914 | 8.114 |
| | (0.006) | (0.026) | (0.014) | (0.205) |
| $D_4$ | −1.526 | 4.439 | 8.933 | 6.618 |
| | (0.005) | (0.030) | (0.014) | (0.208) |
| $D_5$ | −1.566 | 4.782 | 8.935 | 5.139 |
| | (0.006) | (0.030) | (0.013) | (0.246) |
| $D_1 \times D_t^{wknd}$ | 0.202 | 0.201 | 0.028 | 2.414 |
| | (0.002) | (0.015) | (0.005) | (0.346) |
| $D_2 \times D_t^{wknd}$ | 0.205 | 0.241 | 0.044 | 3.416 |
| | (0.002) | (0.010) | (0.005) | (0.278) |
| $D_3 \times D_t^{wknd}$ | 0.210 | 0.255 | 0.054 | 4.480 |
| | (0.002) | (0.009) | (0.005) | (0.299) |
| $D_4 \times D_t^{wknd}$ | 0.218 | 0.308 | 0.055 | 6.147 |
| | (0.002) | (0.009) | (0.006) | (0.320) |
| $D_5 \times D_t^{wknd}$ | 0.229 | 0.256 | 0.039 | 4.888 |
| | (0.002) | (0.010) | (0.005) | (0.281) |
| $D_1 \times ED_{ct}$ | 0.277 | −0.740 | −0.113 | −27.412 |
| | (0.006) | (0.023) | (0.007) | (0.504) |
| $D_2 \times ED_{ct}$ | 0.303 | −0.742 | −0.121 | −29.748 |
| | (0.005) | (0.014) | (0.006) | (0.344) |
| $D_3 \times ED_{ct}$ | 0.326 | −0.775 | −0.130 | −31.531 |
| | (0.005) | (0.013) | (0.005) | (0.351) |
| $D_4 \times ED_{ct}$ | 0.348 | −0.813 | −0.135 | −32.807 |
| | (0.005) | (0.014) | (0.006) | (0.348) |
| $D_5 \times ED_{ct}$ | 0.458 | −1.040 | −0.177 | −36.276 |
| | (0.007) | (0.023) | (0.006) | (0.391) |
| $D_1 \times ED_{ct} \times D_t^{wknd}$ | −0.051 | −0.229 | −0.186 | −6.723 |
| | (0.003) | (0.015) | (0.005) | (0.414) |
| $D_2 \times ED_{ct} \times D_t^{wknd}$ | −0.050 | −0.211 | −0.181 | −8.850 |
| | (0.003) | (0.011) | (0.005) | (0.296) |
| $D_3 \times ED_{ct} \times D_t^{wknd}$ | −0.054 | −0.186 | −0.191 | −9.936 |
| | (0.002) | (0.010) | (0.005) | (0.320) |
| $D_4 \times ED_{ct} \times D_t^{wknd}$ | −0.060 | −0.212 | −0.204 | −11.797 |
| | (0.003) | (0.009) | (0.006) | (0.360) |
| $D_5 \times ED_{ct} \times D_t^{wknd}$ | −0.095 | −0.145 | −0.186 | −11.131 |
| | (0.003) | (0.011) | (0.005) | (0.332) |
| $D_1 \times ED_{ct} \times D_t^{pay}$ | −0.080 | 0.467 | 0.027 | 10.877 |
| | (0.004) | (0.010) | (0.004) | (0.888) |
| $D_2 \times ED_{ct} \times D_t^{pay}$ | −0.087 | 0.468 | 0.017 | 15.443 |
| | (0.004) | (0.008) | (0.003) | (0.592) |
| $D_3 \times ED_{ct} \times D_t^{pay}$ | −0.078 | 0.454 | 0.016 | 15.010 |
| | (0.004) | (0.008) | (0.004) | (0.512) |

(*Continued*)

**Table 2.** (Continued)

| | Complete Home | Device Expos. | Med. Dist. Traveled | Retail & Recreation |
|---|---|---|---|---|
| $D_4 \times ED_{ct} \times D_t^{pay}$ | −0.064 | 0.433 | 0.016 | 15.804 |
| | (0.004) | (0.009) | (0.004) | (0.619) |
| $D_5 \times ED_{ct} \times D_t^{pay}$ | −0.079 | 0.378 | 0.030 | 13.772 |
| | (0.005) | (0.011) | (0.005) | (0.636) |
| $D_1 \times ED_{ct} \times D_t^{pay} \times D_t^{wknd}$ | −0.052 | 0.196 | 0.083 | 2.905 |
| | (0.002) | (0.007) | (0.005) | (0.614) |
| $D_2 \times ED_{ct} \times D_t^{pay} \times D_t^{wknd}$ | −0.061 | 0.160 | 0.068 | 4.048 |
| | (0.002) | (0.005) | (0.004) | (0.333) |
| $D_3 \times ED_{ct} \times D_t^{pay} \times D_t^{wknd}$ | −0.066 | 0.120 | 0.072 | 3.594 |
| | (0.002) | (0.005) | (0.005) | (0.321) |
| $D_4 \times ED_{ct} \times D_t^{pay} \times D_t^{wknd}$ | −0.076 | 0.121 | 0.068 | 4.072 |
| | (0.002) | (0.005) | (0.004) | (0.327) |
| $D_5 \times ED_{ct} \times D_t^{pay} \times D_t^{wknd}$ | −0.073 | 0.102 | 0.067 | 2.491 |
| | (0.003) | (0.006) | (0.004) | (0.309) |

Overall, in line with the findings in Weill et al. (2020) [5], our regression results show that the adoption of social distancing habits following states' emergency declarations was substantial and strongly differentiated by income level, but that stimulus payments worsened social distancing behavior, especially on weekends. These results are unsurprising given that the stimulus payment is distributed on Friday, just in time for recipients to go shopping on the weekend. The results are consistent with the work of Yang, Choe, and Martell (2020) [8] that stimulus payment has a significant short-term effect of boosting spending. Furthermore, although social distancing habits generally tended to lapse more for lower-income quintiles, when looking specifically at the retail activity it was in fact the middle-income quintiles that showed the greatest increase. This can be explained by the result in Coibon, Gorodnichenko, and Weber (2020) [6], with the lowest income individuals using their stimulus checks to pay down debt and the highest income individuals saving theirs, leaving the middle-income individuals most likely to actually spend the stimulus payment at retail.

## Robustness study

Our previous model (1) adopts the state emergency declaration as the social distancing policy measure. However, the emergency declaration can be regarded as a *macro* or rough proxy. In this section, we consider a *micro* level social distancing policy measure using the data of closure and reopening of restaurants, gyms, movie theaters, and bars provided by the COVID-19 U.S. State Policy (CUSP) database (The data is provided publicly at https://statepolicies.com/). Fig 3 summarizes the closing periods of restaurants, gyms, movie theaters, and bars in different states in our sample period. The specific closing and reopening dates of various businesses are slightly different. We choose the earliest closing date of restaurants, gyms, movie theaters, and bars and the latest reopening date to proxy the overall closing period of business in each state. We can see in the figure that there is substantial variation in the *micro level* social distancing policy measure compared to the uniform policy that we might expect from the emergency declaration. Indeed, the reopening of businesses began as early as May 1st in North Dakota and Utah, although all were still in lockdown as of the commencement of stimulus payments in April.

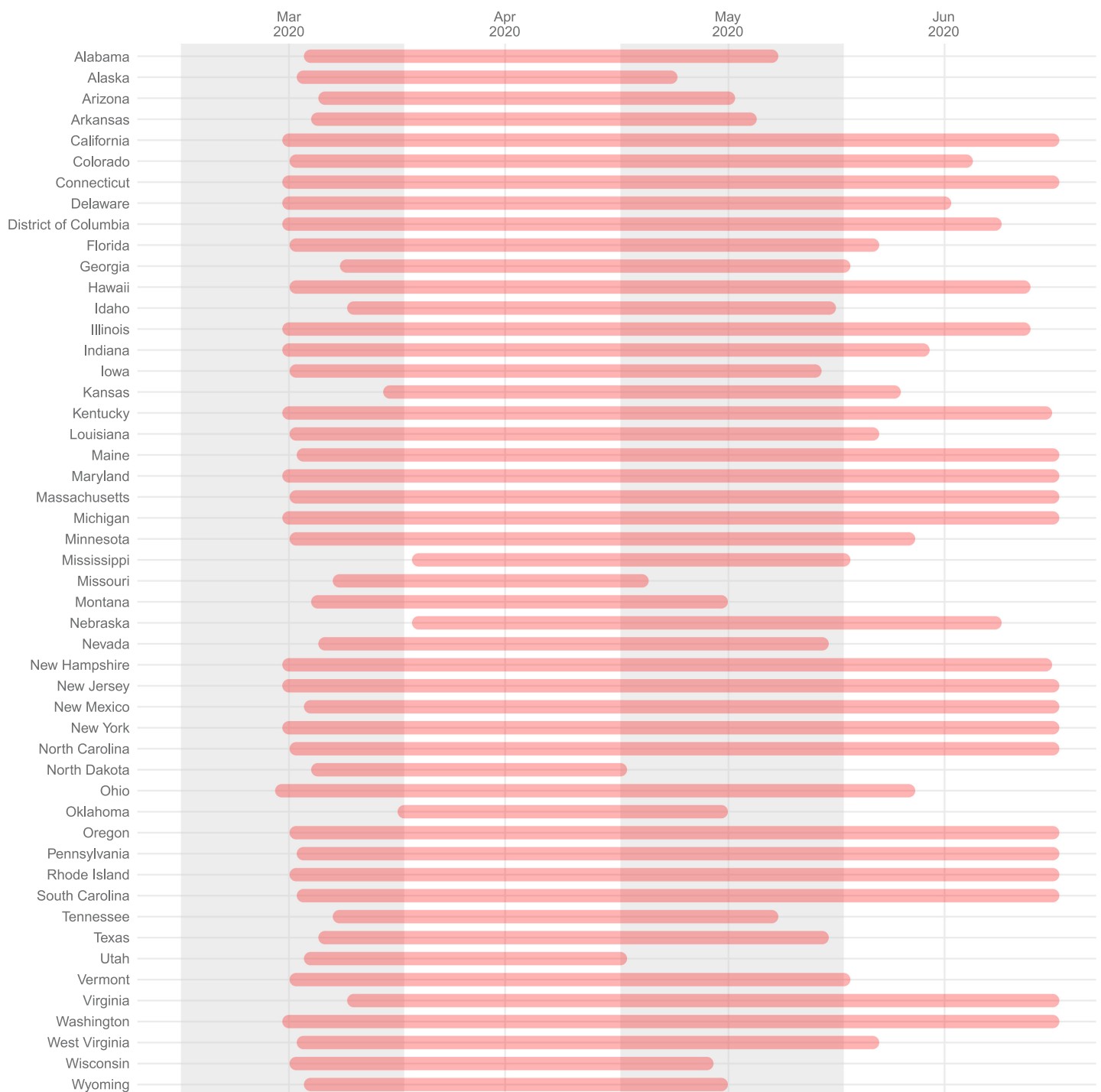

**Fig 3. Timeline of closures of restaurants, gyms, movie theaters, and bars in different states in our sample period.**

**Table 3. The regression models of four separate mobility measures on income quintiles ($D_q$), weekend effect ($D_t^{wknd}$), business closure ($BUSIN_{ct}$), and the stimulus payment period ($D_t^{pay}$).** Standard errors are given in parentheses.

| | Complete Home | Device Expos. | Med. Dist. Traveled | Retail & Recreation |
|---|---|---|---|---|
| $D_1$ | −1.389 | 4.357 | 8.883 | 2.075 |
| | (0.014) | (0.025) | (0.016) | (0.623) |
| $D_2$ | −1.497 | 4.431 | 8.943 | 4.147 |
| | (0.007) | (0.021) | (0.013) | (0.459) |
| $D_3$ | −1.465 | 4.457 | 8.900 | 2.882 |
| | (0.006) | (0.025) | (0.014) | (0.468) |
| $D_4$ | −1.454 | 4.371 | 8.899 | 1.768 |
| | (0.006) | (0.029) | (0.015) | (0.557) |
| $D_5$ | −1.452 | 4.604 | 8.896 | −2.212 |
| | (0.007) | (0.031) | (0.013) | (0.639) |
| $D_1 \times D_t^{wknd}$ | 0.156 | 0.173 | −0.007 | 0.719 |
| | (0.002) | (0.011) | (0.005) | (0.309) |
| $D_2 \times D_t^{wknd}$ | 0.171 | 0.209 | 0.025 | 0.879 |
| | (0.002) | (0.008) | (0.005) | (0.247) |
| $D_3 \times D_t^{wknd}$ | 0.177 | 0.226 | 0.032 | 1.498 |
| | (0.002) | (0.007) | (0.005) | (0.254) |
| $D_4 \times D_t^{wknd}$ | 0.182 | 0.270 | 0.034 | 1.868 |
| | (0.002) | (0.008) | (0.005) | (0.306) |
| $D_5 \times D_t^{wknd}$ | 0.181 | 0.221 | 0.011 | 0.143 |
| | (0.003) | (0.008) | (0.005) | (0.246) |
| $D_1 \times BUSIN_{ct}$ | 0.149 | −0.782 | −0.005 | −28.665 |
| | (0.014) | (0.023) | (0.015) | (0.747) |
| $D_2 \times BUSIN_{ct}$ | 0.324 | −0.840 | −0.133 | −32.730 |
| | (0.005) | (0.017) | (0.009) | (0.552) |
| $D_3 \times BUSIN_{ct}$ | 0.351 | −0.859 | −0.142 | −34.627 |
| | (0.006) | (0.017) | (0.008) | (0.549) |
| $D_4 \times BUSIN_{ct}$ | 0.371 | −0.882 | −0.115 | −36.460 |
| | (0.006) | (0.019) | (0.011) | (0.614) |
| $D_5 \times BUSIN_{ct}$ | 0.450 | −1.046 | −0.173 | −37.247 |
| | (0.009) | (0.028) | (0.010) | (0.698) |
| $D_1 \times BUSIN_{ct} \times D_t^{wknd}$ | 0.009 | −0.278 | −0.215 | −7.760 |
| | (0.003) | (0.013) | (0.006) | (0.441) |
| $D_2 \times BUSIN_{ct} \times D_t^{wknd}$ | −0.012 | −0.257 | −0.217 | −9.313 |
| | (0.003) | (0.010) | (0.006) | (0.322) |
| $D_3 \times BUSIN_{ct} \times D_t^{wknd}$ | −0.016 | −0.243 | −0.230 | −10.590 |
| | (0.003) | (0.009) | (0.006) | (0.306) |
| $D_4 \times BUSIN_{ct} \times D_t^{wknd}$ | −0.025 | −0.278 | −0.251 | −11.171 |
| | (0.003) | (0.009) | (0.006) | (0.386) |
| $D_5 \times BUSIN_{ct} \times D_t^{wknd}$ | −0.052 | −0.212 | −0.209 | −9.890 |
| | (0.004) | (0.009) | (0.006) | (0.312) |
| $D_1 \times BUSIN_{ct} \times D_t^{pay}$ | −0.103 | 0.463 | 0.053 | 15.710 |
| | (0.005) | (0.012) | (0.005) | (0.885) |
| $D_2 \times BUSIN_{ct} \times D_t^{pay}$ | −0.114 | 0.439 | 0.045 | 19.277 |
| | (0.003) | (0.010) | (0.004) | (0.581) |
| $D_3 \times BUSIN_{ct} \times D_t^{pay}$ | −0.123 | 0.459 | 0.039 | 20.323 |
| | (0.003) | (0.010) | (0.005) | (0.515) |

*(Continued)*

**Table 3.** (Continued)

| | Complete Home | Device Expos. | Med. Dist. Traveled | Retail & Recreation |
|---|---|---|---|---|
| $D_4 \times BUSIN_{ct} \times D_t^{pay}$ | −0.119 | 0.473 | 0.037 | 19.886 |
| | (0.004) | (0.011) | (0.005) | (0.632) |
| $D_5 \times BUSIN_{ct} \times D_t^{pay}$ | −0.124 | 0.464 | 0.064 | 17.186 |
| | (0.005) | (0.011) | (0.005) | (0.591) |
| $D_1 \times BUSIN_{ct} \times D_t^{pay} \times D_t^{wknd}$ | −0.067 | 0.239 | 0.146 | 5.162 |
| | (0.002) | (0.009) | (0.006) | (0.648) |
| $D_2 \times BUSIN_{ct} \times D_t^{pay} \times D_t^{wknd}$ | −0.063 | 0.209 | 0.122 | 6.831 |
| | (0.002) | (0.007) | (0.005) | (0.362) |
| $D_3 \times BUSIN_{ct} \times D_t^{pay} \times D_t^{wknd}$ | −0.071 | 0.182 | 0.131 | 7.148 |
| | (0.002) | (0.006) | (0.005) | (0.318) |
| $D_4 \times BUSIN_{ct} \times D_t^{pay} \times D_t^{wknd}$ | −0.076 | 0.197 | 0.134 | 7.587 |
| | (0.002) | (0.006) | (0.005) | (0.333) |
| $D_5 \times BUSIN_{ct} \times D_t^{pay} \times D_t^{wknd}$ | −0.074 | 0.180 | 0.110 | 6.001 |
| | (0.002) | (0.007) | (0.005) | (0.321) |

The corresponding regression model is thus

$$
\begin{aligned}
Y_{ct} &= \sum_{q \in Q} c_q \cdot D_q + \sum_{q \in Q} \alpha_q \cdot D_q \times D_t^{wknd} + \sum_{q \in Q} \beta_q \cdot D_q \times BUSIN_{ct} \\
&+ \sum_{q \in Q} \gamma_q \cdot D_q \times BUSIN_{ct} \times D_t^{wknd} + \sum_{q \in Q} \theta_q \cdot D_q \times BUSIN_{ct} \times D_t^{pay} \\
&+ \sum_{q \in Q} \eta_q \cdot D_q \times BUSIN_{ct} \times D_t^{pay} \times D_t^{wknd} + \zeta \cdot X_{ct} + \epsilon_{ct},
\end{aligned}
\tag{2}
$$

where the variable $ED_{ct}$ in (1) is replaced by $BUSIN_{ct}$—a dummy variable indicating the closing period of business in county $c$.

The results using the business closure policy ($BUSIN_{ct}$) in Table 3 are quite robust to those using the state emergency declaration ($ED_{ct}$). Overall, we found that after the business closure, social distancing increases significantly for all income groups ($D_q \times BUSIN_{ct}$). In particular, the social distancing gradually increases from in income-level group 1 ($D_1 \times BUSIN_{ct}$) to group 5 ($D_5 \times BUSIN_{ct}$). It is also expected that the business closure increased social distancing behavior on weekends ($D_q \times BUSIN_{ct} \times D_t^{wknd}$). Similar to the results in Table 2, the stimulus payment significantly reversed the social distancing behavior. The results show that after the stimulus payment on April 17th, social distancing behavior loosens for all income groups ($D_q \times BUSIN_{ct} \times D_t^{pay}$); in particular, the response to stimulus payment on weekends is far worse than on weekdays for all income groups ($D_q \times BUSIN_{ct} \times D_t^{pay} \times D_t^{wknd}$). Overall, the results using business closure as the distancing policy measure reaffirm the findings of the previous model.

## Conclusion

In our study, we found that the stimulus payments in every instance led to reduced social distancing practices, with the obvious conclusion that receiving the money encouraged individuals to go out and spend it right away. While some of this could be explained by the general tendency to shop more on weekends, even controlling for this we found that the stimulus

payments still led to more retail activity and thus weaker social distancing. While this may have had the intended economic effect of stimulating consumer spending, it raises the critical question of whether this policy contributed significantly to further spreading the virus and extending the pandemic, which would likely have a negative impact on economic performance that far outweighed the effect of the stimulus itself, something which we hope to be able to answer through future investigation. We also emphasize that changes in human mobility in response to monetary policy can vary across countries which adopted different financial relief measures such as tax deferrals and public guarantees to direct grants and coupons. It would be interesting for future research to compare the effects of different financial relief measures across countries during the COVID-19 pandemic.

## Author Contributions

**Conceptualization:** Cody Yu-Ling Hsiao, Stanley Iat-Meng Ko, Nan Zhou.

**Data curation:** Stanley Iat-Meng Ko.

**Formal analysis:** Nan Zhou.

**Investigation:** Cody Yu-Ling Hsiao, Stanley Iat-Meng Ko, Nan Zhou.

**Methodology:** Cody Yu-Ling Hsiao, Stanley Iat-Meng Ko, Nan Zhou.

**Visualization:** Stanley Iat-Meng Ko.

**Writing – original draft:** Cody Yu-Ling Hsiao, Nan Zhou.

**Writing – review & editing:** Nan Zhou.

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
