## [Decision Letter · Decision Letter 0]

5 Apr 2022

PGPH-D-22-00258

Financial relief policy and social distancing during the COVID-19 pandemic

Dear Dr. Zhou,

Thank you for submitting your manuscript to PLOS Global Public Health. After careful consideration, we feel that it has merit but does not fully meet PLOS Global Public Health’s publication criteria as it currently stands. Therefore, we invite you to submit a revised version of the manuscript that addresses the points raised during the review process.

We look forward to receiving your revised manuscript.

Kind regards,

Ener Cagri Dinleyici

Academic Editor

Journal Requirements:

1. Your co-author, Cody Hsiao (ylhsiao@must.edu.mo), has not confirmed authorship of the manuscript. We have resent them the authorship confirmation email; however please check that the above email address for them is correct and follow up personally to ensure they confirm. Please note that we cannot pass your manuscript to Production until we have received confirmations from all co-authors. 

Just in case your co-authors are having difficulty confirming their authorship, you may advise them to send us an email at globalpubhealth@plos.org and we will confirm their authorship on the authors' behalf.

2. Please amend your Financial Disclosure statement. If you did not receive any funding for this study, please simply state: “The authors received no specific funding for this work.”

3. Please update your Competing Interests statement. If you have no competing interests to declare, please state: “The authors have declared that no competing interests exist.”

4. We ask that a manuscript source file is provided at Revision. Please upload your manuscript file as a .doc, .docx, .rtf or .tex. If you are providing a .tex file, please upload it under the item type ‘LaTeX Source File’ and leave your .pdf version as the item type ‘Manuscript’.

5. Please provide separate figure files in .tif or .eps format only and ensure that all files are under our size limit of 20MB.

Additional Editor Comments (if provided):

The reviewers recommend Major Revision at this time. Please note that submitting a revision of your manuscript does not guarantee eventual acceptance, and that your revision will be subject to re-review by the referees before a decision is rendered.

Reviewers' comments:

Reviewer's Responses to Questions

**Comments to the Author**

1. Does this manuscript meet PLOS Global Public Health’s publication criteria? Is the manuscript technically sound, and do the data support the conclusions? The manuscript must describe methodologically and ethically rigorous research with conclusions that are appropriately drawn based on the data presented.

Reviewer #1: Yes

Reviewer #2: Yes

2. Has the statistical analysis been performed appropriately and rigorously?

Reviewer #1: No

Reviewer #2: Yes

3. Have the authors made all data underlying the findings in their manuscript fully available (please refer to the Data Availability Statement at the start of the manuscript PDF file)?

Reviewer #1: Yes

Reviewer #2: Yes

4. Is the manuscript presented in an intelligible fashion and written in standard English?

Reviewer #1: Yes

Reviewer #2: Yes

5. Review Comments to the Author

Reviewer #1: • Although the paper examines an interesting research question, the authors fail to discuss why they run different regressions without controlling many other independent variables, potentially impacting mobility (social distancing). For example, during the COVID period, we know many other stringency policies are implemented by policymakers, potentially impacting social distancing. In other words, during the analysis period of the paper, many different countries implemented many control policies outside of financial relief policy. In this context, how can the paper say that the change in mobility is caused by financial relief policy without controlling the potancially effect of other policy implementations such as international travel controls, cancellation of public events, closing of public transportation, the closing of schools, closing of workplaces, restrictions on internal movements, etc. (please see Oxford COVID-19 Government Response Tracker database, Thomas et al., 2020). For example, the policymakers may have applied less stringent policies in the relevant period, and mobility may have increased due to this implementation.

• The author should express the difference between this study from the common literature (see Mendolia et. Al., 2020).

• The article must support the reliability of empirical findings by adding different control variables to the model. In other words, the paper needs to present a robustness check of their empirical findings.

• When results are interpreted, there is a need to draw out from the literature arguments to support or dispute the findings. There is an inadequate discussion of the results. As there is little discussion of the implications of the results, the reader finds it hard to interpret the results. In other words, it is necessary to identify the wider implications of this set of findings. In the absence of a satisfactory discussion, the reader is left very much in the dark as to the specifics of the implications of the results.

• A possible 'limitation' should be stressed is that the USA may differ from other countries. If other researchers try to replicate this research, results may vary due to different countries. The Reviewer believes that it is important to stress this point.

References

Hale, T., Webster, S., Petherick, A., Phillips, T., & Kira, B. (2020). Oxford COVID-19 government response tracker (OxCGRT). Last updated, 8, 30.

Mendolia, S., Stavrunova, O., & Yerokhin, O. (2021). Determinants of the community mobility during the COVID-19 epidemic: The role of government regulations and information. Journal of Economic Behavior & Organization, 184, 199-231

Reviewer #2: Though paper tries to establish a negative relationship between economic stimulus and social distancing, however it is difficult to conclude whether economic support actually increased the social distancing at least for two reasons. First, it might just be natural to go for shopping on the weekend. We have this trend from centuries. Second, individuals might have not gone for those trips in the anticipation of economic support for which otherwise they might have gone. This information is subjective and not shown in the data. Auhors may further recognise this as a shortcoming in the paper.

6. PLOS authors have the option to publish the peer review history of their article (what does this mean?). If published, this will include your full peer review and any attached files.

**Do you want your identity to be public for this peer review?** For information about this choice, including consent withdrawal, please see our Privacy Policy.

Reviewer #1: No

Reviewer #2: **Yes: **Badar Nadeem Ashraf

---

## [Decision Letter · Decision Letter 1]

31 May 2022

Financial relief policy and social distancing during the COVID-19 pandemic

PGPH-D-22-00258R1

Dear Dr Zhou,

We are pleased to inform you that your manuscript 'Financial relief policy and social distancing during the COVID-19 pandemic' has been provisionally accepted for publication in PLOS Global Public Health.

Best regards,

Ener Cagri Dinleyici

Academic Editor

Reviewer Comments (if any, and for reference):

Reviewer's Responses to Questions

**Comments to the Author**

1. If the authors have adequately addressed your comments raised in a previous round of review and you feel that this manuscript is now acceptable for publication, you may indicate that here to bypass the “Comments to the Author” section, enter your conflict of interest statement in the “Confidential to Editor” section, and submit your "Accept" recommendation.

Reviewer #1: All comments have been addressed

2. Does this manuscript meet PLOS Global Public Health’s publication criteria? Is the manuscript technically sound, and do the data support the conclusions? The manuscript must describe methodologically and ethically rigorous research with conclusions that are appropriately drawn based on the data presented.

Reviewer #1: Yes

3. Has the statistical analysis been performed appropriately and rigorously?

Reviewer #1: Yes

4. Have the authors made all data underlying the findings in their manuscript fully available (please refer to the Data Availability Statement at the start of the manuscript PDF file)?

Reviewer #1: Yes

5. Is the manuscript presented in an intelligible fashion and written in standard English?

Reviewer #1: Yes

6. Review Comments to the Author

Reviewer #1: The authors of the article have addressed more or less deeply the points suggested in my previous report. I have no further suggestions.

7. PLOS authors have the option to publish the peer review history of their article (what does this mean?). If published, this will include your full peer review and any attached files.

**Do you want your identity to be public for this peer review?** For information about this choice, including consent withdrawal, please see our Privacy Policy.

Reviewer #1: No
